# An NLO-Matched Initial and Final State Parton Shower on a GPU

**Michael H. Seymour**[1,2] **and Siddharth Sule**[1⋆]

1. Department of Physics and Astronomy,
The University of Manchester, M13 9PL, United Kingdom
2. School of Physics and Astronomy,
University of Southampton, SO17 1BJ, United Kingdom

⋆ siddharth.sule@manchester.ac.uk

## Abstract

Recent developments have demonstrated the potential for high simulation speeds and reduced energy consumption by porting Monte Carlo Event Generators to GPUs. We release version 2 of the CUDA C++ parton shower event generator GAPS, which can simulate initial and final state emissions on a GPU and is capable of hard-process matching. As before, we accompany the generator with a near-identical C++ generator to run simulations on single-core and multi-core CPUs. Using these programs, we simulate NLO Z production at the LHC and demonstrate that the speed and energy consumption of an NVIDIA V100 GPU are on par with a 96-core cluster composed of two Intel Xeon Gold 5220R Processors, providing a potential alternative to cluster computing.

# 1   Introduction

The simulation of particle physics events at colliders like the LHC is becoming increasingly accurate and precise. However, these simulations become increasingly computationally expensive and introduce data bottlenecks. The last few years have seen active implementation of GPU-accelerated solutions, ranging from amplitudes and loops [1–4] to PDF evaluation [5, 6], leading to parton-level Monte Carlo Event Generators [7–9]. Connecting these hard processes to particle-level events requires simulating the parton shower, multi-parton interactions and hadronisation. The only current option is to output the data and read it into a CPU-only event generator, which is a significant bottleneck. We work to solve this issue by porting these components to the GPU, building towards a complete GPU Event Generator.

GPUs are traditionally designed with the Single Instruction Multiple Thread Paradigm, where each core of the GPU runs the same command. However, due to the nature of event generation, where each event can follow a different trajectory, divergence can arise, where different cores need to perform different tasks. A simple example is an if–else statement that results in half of the GPU running command A and the other half running command B. Conventionally, this is implemented by the GPU issuing command A, which is executed by half the cores while the other half wait, before executing command B while the first half wait. The size of the divergence is significantly reduced in modern GPUs, which are split into smaller, independently managed groups called warps [10, ch. 7]. Algorithms can be restructured and adjusted to minimise this divergence, but at the cost of making it harder for developers to make changes and for newer developers to learn. In our approach, we port the parton shower veto algorithm to the GPU with minimal changes. We then evaluate the acceleration obtained and identify bottlenecks for interested developers to improve and innovate. This approach was employed in our first work [11], and we continue to follow it. We also release a proof-of-concept CUDA C++ event generator that implements our adjusted algorithm. We additionally provide a complementary C++ event generator that runs the original algorithms to demonstrate the changes needed to port the code. We validate this port by generating identical results from both generators. We also cross-check the physics by comparing the results to state-of-the-art, production-ready event generators. The two generators and all the tools needed to reproduce the results presented in the paper are available in our package called GAPS, a GPU-Amplified Parton Shower [12]. The latest version can be found at

https://gitlab.com/siddharthsule/gaps

In our previous work, we attached a final-state parton shower to the leading-order (LO) hard process $e^+e^- \to Z/\gamma \to q\bar{q}$. Production-level parton showers, as used for studies at the LHC and the Tevatron, can simulate initial-state emissions, i.e. emissions before the hard process, and can connect to next-to-leading-order (NLO) processes, which lead to more accurate simulation results. In this study, we expand on our previous work by simulating initial-state emissions and by connecting the shower to the simple NLO process $pp \to Z$. In Section 2, we describe the updated parton shower veto algorithm for initial state emission. In Section 3, we describe a few computational tricks that can be used to achieve efficiency and speedup without altering the algorithms. The results of these changes are presented in section 4. Here, we cross-check the physics by comparing our results with those from the Herwig Event Generator. We then evaluate the acceleration by comparing the execution time and energy consumption of the CUDA C++ code on the GPU with those of the parallel C++ code on a 96-core cluster, thereby determining whether GPU simulations can be an alternative to CPU cluster simulations. We also profile the code to identify opportunities for improvement in the implementation. The appendices A and B describe the implemented parton shower physics, while appendix C covers the calculation for $pp \to Z$ at NLO for reference.

## 2   Updates to the Parallelised Veto Algorithm

In our parallelised version of the parton shower veto algorithm, we break the algorithm into its steps and run them in parallel for every event on the GPU. We describe the structure of the event as a list of colour–anticolour dipoles. The first step is to generate an emission from each dipole according to a trial distribution, and to find the trial emission from among all the dipoles that occurs at the highest momentum scale: we call this "selecting the winner" emission. The next step is to use the full expression for the emission probability distribution to calculate the probability of retaining this emission. We choose to accept or veto it with this probability. If accepted, the final step is to calculate the full kinematics of the emission. In either case, the algorithm then loops back to generate further trial emissions at lower momentum scales. The evolution terminates when the winner emission is at a scale below the predefined cutoff scale of the shower. Further details can be found in our previous work [11].

The algorithm can be adapted to simulate initial-state radiation by incorporating the effects of parton distribution functions (PDFs), which describe the probability distributions of partons in the incoming proton. These functions guide the backwards evolution of the emitting partons, from the interaction to the parent proton. More details of including PDFs in the veto algorithm are provided in appendix A, and our shower model is detailed in appendix B. The main point here is that the PDFs can only be evaluated after the trial emission has been generated and are valid only for scales above the shower cutoff scale. This necessitates breaking apart the "calculate acceptance probability" step into three substeps. The rest of the veto algorithm is unchanged. An updated illustration of the algorithm is shown in Figure 1.

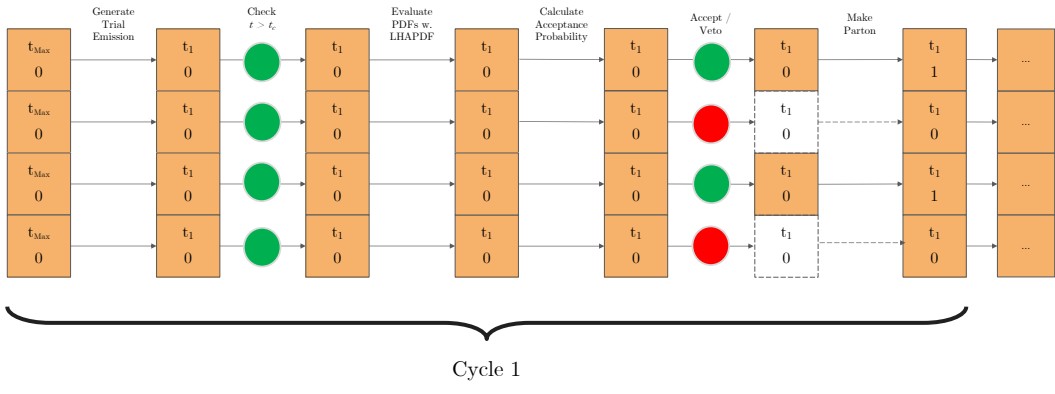

Figure 1: The updated parallelised veto algorithm. The PDF evaluations are required for calculating the emission acceptance probability and are therefore performed before that step.

We incorporate the GPU PDF evaluations offered by LHAPDF in our code [5, 7]. At the time of this report, LHAPDF offers parallelised 2D Interpolation in $(x, Q^2)$ space. However, in simulating processes such as $q\bar{q} \to e^+ e^-$, different events have different flavours of quarks (as well as different $x$ and $Q^2$ values). This issue is further escalated during the shower stage, since the backward evolution is not diagonal and each event can have its own combination of quarks, antiquarks and gluons. As a first solution, our approach was to evaluate $f(\text{flav}, x, Q^2)$ for *ALL* flavours (one flavour at a time), and pick the correct value out of the $N_{ev} \times N_{flav}$ array. Using LHAPDF and our own CUDA Kernels, this method is implemented for Hard Process and Parton Shower. Although cumbersome, this method is intuitive. To test the impact of this approach, we calculated 1,000,000 PDFs for randomised $(\tilde{ij}, i, x, q^2)$ on both CPU and the GPU. Evaluating one case at a time on a single Intel Xeon E5-2620 CPU core [13] took

0.145 seconds, while evaluating all cases in parallel on an NVIDIA V100 GPU [14] took 0.002 seconds, about 70 times faster than the CPU. These results can be reproduced using the file `test/pdf-evaluations.cu` in the repository.

As discussed in our previous work, divergences when processing different events are encountered in this algorithm in several ways. The first arises when generating a trial emission for all dipoles in the event, because different events have different multiplicities. Each event in a warp has to wait until all the events in that warp have generated all trial emissions before they can proceed to the next step of checking the emission scale against the cutoff. The second occurs after the accept/veto step, where events with rejected emissions must wait for events with accepted emissions to generate the new parton before all events take the next step of generating the next trial emission. The last is the simple fact that different events generate different numbers of emissions before the cutoff, so the algorithm cycles through different numbers of times. Events that finish quickly have to wait for events that take longer. Our approach in the previous paper was to test whether, despite these divergences, a reasonable speedup can be obtained on the GPU without altering the veto algorithm, such that generator developers can adapt their code without significantly changing their implementations. We continue with that approach. However, in the next section, we consider whether some small improvements can be made to the algorithm by considering the layout of the events on the GPU.

## 3  Computational Improvements

In this section, we explore the possible improvements and optimisations we can implement without altering the veto algorithm. The first method focuses on preventing unnecessary computations by discarding events that have already completed. The second looks into finding the optimal GPU execution settings for the code.

### 3.1  Partitioning of the Event Record List

As we discussed above and in [11], events that had finished showering had to wait for the unfinished events to shower, resulting in many unused GPU computations. However, the time taken to shower the LEP events considered in [11] was small (1,000,000 events is 0.5s), and any optimisation methods added more time. On the other hand, the time taken to shower LHC events is of the order of minutes (which we will discuss in the following section), and we can add some time to implement optimisation tricks, as it would form a smaller fraction of the operating time. To allow completed events to free up threads so new events can be started, we have employed a technique we call partitioning. At some point during event processing, we reorder the list of events into events with finished and unfinished showers. Then, we run our GPU kernels with the kernel size set to the number of unfinished events, ensuring the GPU performs calculations only for those events. This algorithm is illustrated in Figure 2. As a first attempt, we did this after 1/2, 1/4, 1/8... unfinished events were left, or until only 10 unfinished events were left.

### 3.2  GPU Kernel Tuning

Modern GPUs have thousands of cores. As mentioned before, these cores are bundled into warps. In addition, a collection of warps is supervised together and shares memory, forming a unit called a *streaming multiprocessor*. The GPU is a collection of these multiprocessors [14,16]. It is often difficult for developers to map every problem to the exact number of GPU cores. As a solution, the parallelisation is defined using pseudo-objects called threads and blocks. A

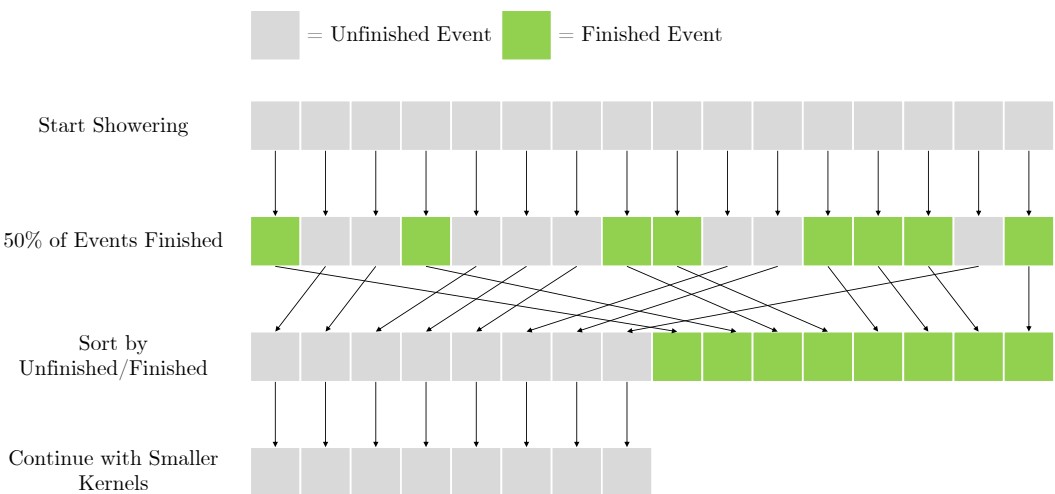

Figure 2: Partitioning the event record list. In this case, $N$ events begin showering. After $N/2$ have finished, the events are partitioned into unfinished events, followed by finished events, which is automated using the `thrust::partition` feature in CUDA [15]. After this partitioning, the kernel is launched with $N/2$ threads, leaving the finished events unaffected. In practice, some events finish showering at the same steps, and the kernels are launched with $N - N_{\mathrm{finished}}$ threads. This version allows us to partition the list of event records at any point in the simulation.

thread is a single calculation corresponding to an element of data and runs on a single core. A block is a group of threads. A block can span multiple warps, but, as a rule, it is allowed to run only on a single streaming multiprocessor. On the other hand, a streaming multiprocessor can run as many blocks of threads as there is space [10, ch. 3,7]. It is helpful to define five useful variables when discussing this,

- $N_{T/B} = N_T$: Number of threads per block

- $N_B$: Number of blocks

- $N_{C/W} = N_C$: Number of cores in a warp

- $N_{W/S} = N_W$: Number of warps in a streaming multiprocessor

- $N_S$: Number of streaming multiprocessors

- $N_{EV}$: Number of Events/Tasks

$N_C$, $N_W$ and $N_S$ are set by the GPU one is running on; the user sets $N_{EV}$; and $N_T$ and $N_B$ can be chosen to optimise the simulation's performance.

A few constraints apply to the selection of $(N_T, N_B)$ for the kernels. Firstly, to ensure that entire warps are used, $N_T$ must be a multiple of $N_C$. This constraint also confines the upper limit of threads per block to be $N_C \times N_W$, corresponding to one block per streaming multiprocessor. Secondly, the number of events a user needs is a free choice, and the code should return the same number. These two constraints can be quantified as

$$N_T \in [N_C, N_C \times N_W], \quad N_B = \mathrm{ceil}\left(\frac{N_{EV}}{N_T}\right). \tag{1}$$

If $N_{EV} > N_S \times N_W \times N_C$, the GPU will process the remaining events as streaming multiprocessors become free. Selecting a value for $N_T$ affects the GPU efficiency and, consequently, the execution time, especially when working with 2D or even 3D kernels. The easiest way to find the optimal value of $N_T$ is to run simulations with different values of $N_T$ and measure their execution metrics. This process is often referred to as *kernel tuning*; optimisation techniques and tools exist for optimising kernels, especially 2D and 3D kernels [17, 18]. As we only employ 1D kernels, our options for $N_T$ are limited to 32, 64, 128, 256, 512, and (if the GPU can handle it) 1024 and 2048. In our code, we set $N_T$ to the same value for all kernels and choose the best value by testing all the mentioned choices. In dedicated scenarios, tuning of the individual kernels can be done, but this was beyond the scope of this report.

# 4   Implementation and Results for LHC at 13 TeV

In this section, we cover the details of our implementation, which is available as GAPS 2. We start with Physics validation for LO+Shower and NLO+Shower. We then examine the impact of the computational improvements on execution time and perform GPU profiling. Finally, we compare the execution time and energy consumption of the GPU generator with those of the CPU generator, which was run on a CPU Cluster.

## 4.1   Validation of Physical Results

We generate the process $p(q)p(\bar{q}) \rightarrow Z/\gamma \rightarrow e^+e^-$ and shower the $q\bar{q}$ pair and all radiated particles. We calculate $Z$ boson observables, such as $m_Z$, $p_T$, $y$, and $\phi$, as well as jet observables using the anti-$k_T$ algorithm [19], similar to the MCZINC and MCZJETS Analyses inside Rivet [20]. As in GAPS 1, the observables were output as Yoda Histograms [21] and plotted with Rivet. The results were compared with simulations performed with the Herwig Event Generator [22], which uses the same parton shower prescription. For context, we ported S Höche's parton shower tutorial for GAPS 1 and were able to generate identical results. However, the implementation in Herwig has a few significant differences that we could not adjust. We discuss further in appendix B, and the further validation studies are included in the documentation. The results are shown in Figure 3. We implemented our own random number generator in both the CPU and GPU versions, allowing them to produce identical results and confirm identical implementations. These results are similar to Herwig for the $Z$ and leading jet observables; small deviations arise for subleading jets and jet multiplicities. These small physics differences are under further investigation, but are not relevant to our present discussion of computational performance.

For NLO Matching, we simplify the process to $p(q)p(\bar{q}) \rightarrow Z$. As we describe in more detail in Appendix C, the calculation produces events with two possible multiplicities and a range of starting scales, which the parallelised veto algorithm can handle by construction. The results were compared to those from MadGraph [23] and OpenLoops [24], matched to Herwig's dipole shower [25], and are shown in Figure 4.

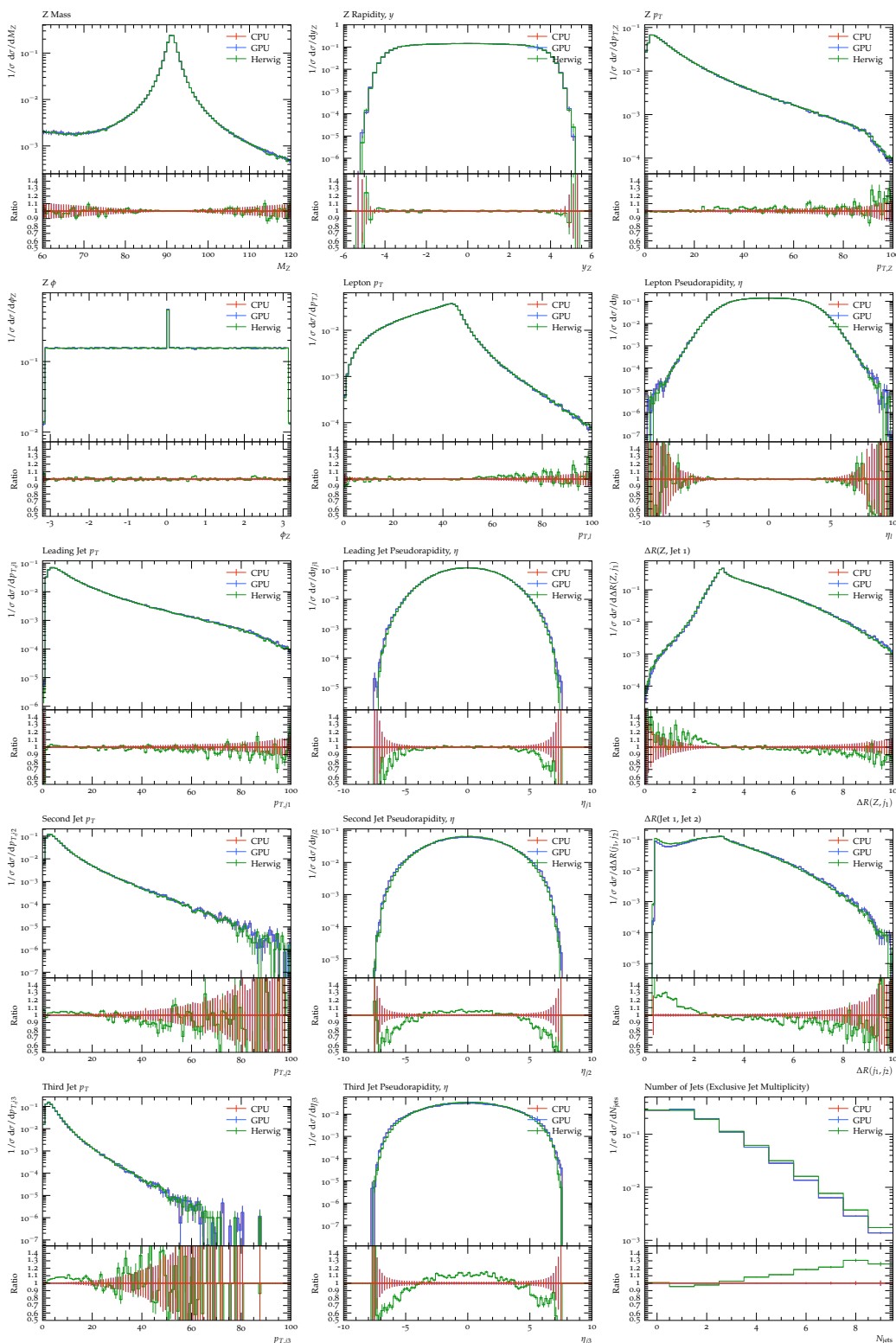

Figure 3: *Z* Observables and Anti-$k_T$ jets produced with $R = 0.4$. The *Z* and lepton observables are fully inclusive, while each jet's $p_T$ distribution is shown when its $|\eta| < 5$, its $\eta$ distribution is shown when its $p_T > 5$ GeV, and the $\Delta R$ and multiplicity distributions are shown when $p_T > 5$ GeV and $|\eta| < 5$. The *Z* and lepton observables agree very well with Herwig, the leading jet pretty well, the second and third jets slightly less well.

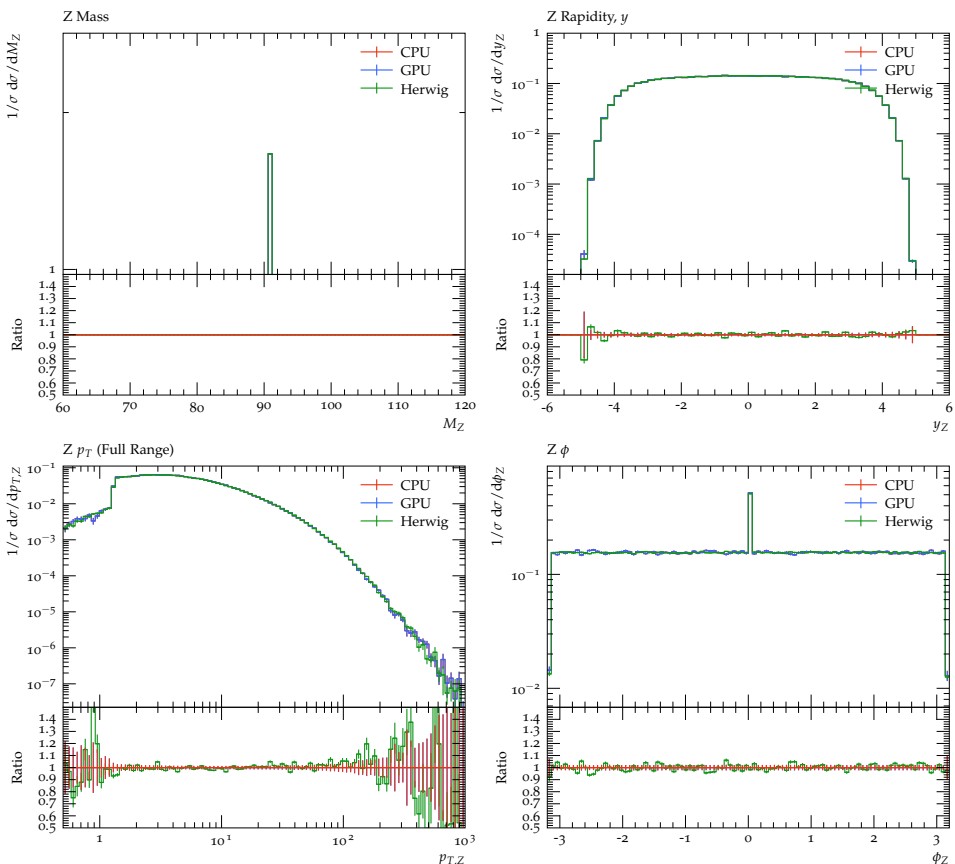

Figure 4: NLO+Shower for the process $pp \rightarrow Z$, where the $Z$ boson is on-shell and stable. Like the LO+Shower case, the $Z$ boson observables are in agreement. The jet observables also contained the same deviations and are omitted here.

## 4.2 GPU Profiling and Impact of Computational Improvements

Similar to our previous work, we used the NVIDIA V100 [14], which has 32 cores per warp and 64 warps per streaming multiprocessor. We used the NLO+Shower setting for all the profiling and execution time analyses below. We focus on simulation results for 10,000, 100,000 and 1,000,000 events. These are the number of events commonly used by users and developers, either to check minor results or perform complete analyses. For complicated NLO events, users may exceed 1,000,000 events to obtain convergence. However, this is not needed for the process we simulated, shown by the results.

As a starting point, the execution time contributed by NLO, Shower and Observables is shown in table 1. These measurements are averaged over ten runs and were run without event partitioning and with $N_T = 256$. The results in the table show that the shower contribution is orders of magnitude larger than the matrix element and observables contribution. As our NLO computation is analytical, the calculation involves evaluating formulae and PDFs in parallel and must be performed once per event. A general-purpose tool like PEPPER or Madgraph [7, 8] would have to generate the matrix element for a more complex process numerically and perform grid integration of the phase space, which would take significantly longer. The computation of observables is very similar to our previous work and requires the same amount of time.

| Component | $10^4$ Events (s) | $10^5$ Events (s) | $10^6$ Events (s) |
|---|---|---|---|
| NLO | 0.04 | 0.05 | 0.07 |
| Shower | 4.21 | 11.16 | 72.94 |
| Observables | 0.06 | 0.08 | 0.24 |
| **Total** | **4.31** | **11.29** | **73.25** |

Table 1: Time taken for each component for different numbers of events, averaged over ten runs. The shower takes much longer than Matrix Element and Observables in our case, so we can approximate the total execution time as the shower execution time plus order 1% inclusions.

The order in which the two improvements are tested can bring about biases and uncertainty. If the partitioning was tested first, which value of $N_T$ should be used? If the kernel was tuned first, should partitioning be on or off? To avoid these biases, we ran the kernel tune with both partitioning on and off. The kernel tuning was performed by running the GPU generator for 10,000, 100,000 and 1,000,000 events, with different values of $N_T$. These values were identical for all kernels in the code; only the external PDF evaluation Kernel from LHAPDF had $N_T = 256$, a fixed setting in their package. The results, averaged over ten runs, are shown in Figure 5. For context, we plot the total time and not the parton showering time. However, as shown above, the matrix element and observables account for only a tiny fraction of the total time. In summary, the results show that partitioning is beneficial for 1,000,000 events, without a strong dependence on $N_T$, but with slightly better performance for $N_T = 128$ than for higher values, for the V100.

After analysing both upgrades, the GPU generator was profiled using Nvidia NSight Systems [26]. The results are shown in table 2. The profiling shows that generating the trial emissions takes almost all of the simulation time. Simulating LHC events generates multiple cascades due to initial-state radiation, resulting in a higher multiplicity than LEP events. Hence, there are many more possible emissions in every cycle, and all of them must be generated before selecting the winning or trial emission. As we aim to evaluate the possibility of running a shower on a GPU without altering the algorithm itself, we choose to accept this result as it is. However, a proposed improvement to this is to use 2D kernels, where the possible emission is generated for every particle in every event, and atomic operations like `atomicMax` [10, ch. 10] used to parallelise the selection of the winning emission.

## 4.3  Comparing a GPU and a CPU Cluster

In our previous work, we calculated the simulation execution time on a CPU+GPU setup and on a single CPU core to estimate how many CPU cores the GPU could replace. As a comparative study, we have used the same methodology and hardware (NVIDIA V100 [14] and Intel Xeon E5-2620 [13]) for our LHC simulation at NLO. For 1,000,000 events, the CPU generator took approximately one hour. The NLO and observable calculations again were a small contribution, taking about 25 seconds in total. The GPU generator, with event partitioning and $N_T = 128$, took about 60 seconds, yielding a speedup of 60. As a back-of-the-envelope calculation, if we assume a power consumption of 5W per core and 250W for the GPU, as before, the total power consumption of 60 cores is 300W, while the GPU and a CPU core consume 255W.

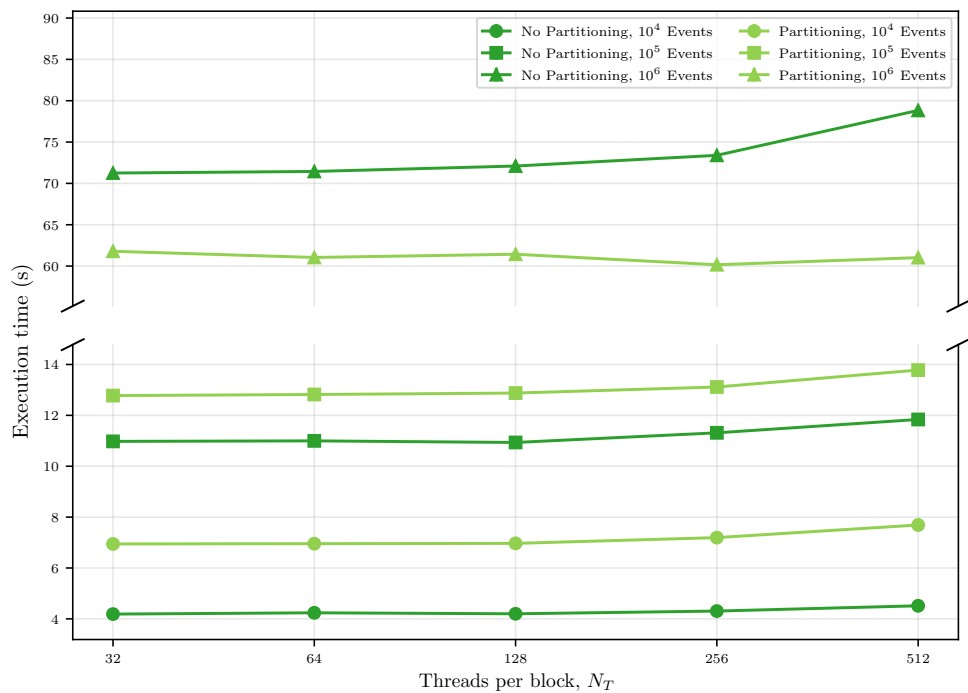

Figure 5: Kernel Tuning Results, with partitioning on and off. For small $N_{EV}$, there is negligible improvement due to the partitioning, and the partitioning even increases the execution time. However, for larger $N_{EV}$ it starts reducing the execution time, offering a 20% reduction in execution time for 1,000,000 events. In terms of kernel tuning, execution times increase slightly with $N_T$ for 10,000 and 100,000 events, but decrease slightly with $N_T$ for 1,000,000 events. The best choice overall is to use $N_T = 128$.

| Name | Instances | Total Time (ns) | Time (%) |
|---|---|---|---|
| Generate the Trial Emission | 1815 | 52,303,264,877.0 | 87.0 |
| PDF (11 Evaluations + Selection) | 3634 | 3,270,274,078.0 | 5.5 |
| *Event Record Partitioning* | 16 | 1,459,083,279.0 | 2.4 |
| Vetoing Process | 1815 | 1,165,600,387.0 | 1.9 |
| Setup PDF Ratio Calculation | 1815 | 578,489,312.0 | 1.0 |
| Perform the Emission | 1815 | 418,441,403.0 | 0.7 |
| *Device Prep* | 1 | 256,456,238.0 | 0.4 |
| Check for Max Particles Per Event | 1815 | 252,243,671.0 | 0.4 |
| Checking Shower Cutoff | 1815 | 229,891,728.0 | 0.4 |
| Anti-$k_T$ Jet Clustering | 1 | 115,152,470.0 | 0.2 |
| Fill Histograms | 1 | 35,312,919.0 | 0.1 |

Table 2: Profiling Results, with event partitioning and $N_T = 128$. Similarly to our previous work, generating the trial emission takes the majority of the time. However, this time around, it accounts for almost all of the simulation time. This is followed by PDF Evaluations, which need to be run 11 times for every flavour, and LHAPDF divides the number of required evaluations into two kernels; the number of evaluations needed is divided by $N_T$, and the remainder is run using another kernel. The LO/NLO hard process requires two/six evaluations, and the shower requires two evaluations per cycle.

However, in this report, we propose a more robust evaluation technique and metrics to study performance. The central issue in comparing a GPU and a CPU generator is to ensure a fair comparison: a GPU is fast, but it also consumes orders of magnitude more power than a CPU core. Additionally, parallel execution on multiple CPUs does not guarantee a $1/N_{\text{Cores}}$ speedup. For this study, we adjusted the CPU generator to run in parallel on multiple cores. As a better test metric, we measured the execution time and the total energy consumption for NLO+Shower on the GPU generator on an Nvidia V100 [14], and the CPU generator on a 96-core unit with two Intel Xeon Gold 5220R Processors [27]. This was done using the CodeCarbon profiling package, which also calculates the emissions of running code [28, 29]. The results, averaged over ten runs, are shown in Figure 6, and demonstrate similar results for the GPU and the CPU Cluster. The measurements do not account for additional power consumed to host the GPU and the CPU Cluster, and the extra processing required to merge the Yoda histograms from the CPU Cluster. The results suggest that the GPU shower, implemented without optimising the algorithm itself, produces results quickly and efficiently and serves as an alternative for users and developers who may have access to a GPU rather than a computer cluster. As GPU clusters become more widespread, this shows that they could be, at least, an alternative to CPU clusters for running event generators, if not somewhat advantageous. Furthermore, individual researchers who may not have access to clusters, but have gaming GPUs on their laptops and small desktops can use these codes locally. For example, GAPS is written with *CUDA Compute Capability 7*, which covers all data center and gaming GPUs released by NVIDIA in the last five years [30].

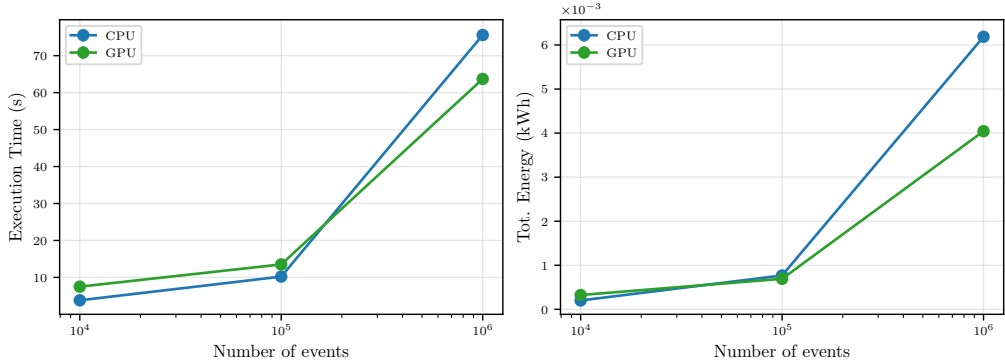

Figure 6: Execution Time, Average power consumption and total energy consumption for CPU Cluster and GPU, at different energies. The execution time is similar for both, increasing with the number of events. The average power consumption is about the same for both, which corresponds to their total energy consumption being similar as well.

## 5 Concluding Remarks and Outlook

The results show that the GPU takes a comparable amount of time and energy to a large computer cluster, showering 1,000,000 NLO events in about a minute. GPU Profiling also shows that selecting the winner emission consumes almost all the processing time during the shower, and 2D kernels may offer an opportunity to distribute the workload better. Still, thanks to matching, the implementation successfully solves the problem of writing and reading in NLO results. It provides a proof of concept for implementation in a future GPU Event Generator.

While this closes the book on the shower, the questions of MPI and hadronisation remain unanswered. Computationally, this is feasible: we have demonstrated event-wide boosts/rescalings and clustering of particles on GPUs, which cover the required calculations for these components. However, only an implementation can answer whether this will result in low execution time and energy consumption. *Adventures await!*

## Acknowledgements

The authors acknowledge using S. Höche's "Introduction to Parton Showers and Matching" tutorial as the starting point for GAPS. The authors use LHAPDF, Yoda, and Rivet in their code, and especially thank M. Knobbe for assistance with using LHAPDF on GPUs. The authors thank S. Plätzer for useful discussion about Dipole Showers and Matching. The authors use Herwig to validate results and thank M.R. Masouminia for assistance with generating them. The authors would like to thank the University of Manchester for access to the Noether Computer Cluster, and thank C. Doglioni and the Blackett Support team for useful discussions.

**Funding information**    SS would like to thank the UK Science and Technology Facilities Council (STFC) for the award of a studentship. MS also acknowledges STFC's support through grants ST/T001038/1 and ST/X00077X/1.

## A    Appendix: Initial State Parton Showers

This section covers how the parton shower veto algorithm is adjusted for initial state radiation. For readers new to the veto algorithm, useful starting points include books [31,32] and review papers [33,34]. Additionally, an introduction to parton shower simulation is provided in our previous work [11].

### A.1    Veto Algorithm with Initial State Radiation

Producing forward evolution of quarks from their parent protons would lead to fewer events with the hard process of interest. Instead, parton showers are designed to evolve *backwards*, that is, from the hard process to the parent proton. Formulating initial state radiation in this approach involves augmenting the final state radiation formulae with the constraint that the emitting particle must reconnect to the proton. Hence, the emissions are "guided" back to the proton, using parton distribution functions or PDFs [31, ch. 5]. This is convenient as our emissions are also based on DGLAP evolution formulae, which are used to evolve PDFs [35–37]. The probability for an initial state emission is calculated by inserting a ratio of parton distributions of the emitting parton before and after the emission[1] ,

$$
\begin{aligned}
\mathcal{P}_{\widetilde{ij}\to i,j}(t,T;\eta_{\widetilde{ij}}) &= 1 - \exp\left[-\int_t^T \frac{\mathrm{d}\hat{t}}{\hat{t}} \int_{z_-}^{z_+} \mathrm{d}\hat{z}\, \frac{\alpha_s(\hat{t},\hat{z})}{2\pi} P_{\widetilde{ij}\to i,j}(\hat{z}) \frac{f(i,\eta_{\widetilde{ij}}/\hat{z},\hat{t})}{f(\widetilde{ij},\eta_{\widetilde{ij}},\hat{t})}\right] \\
&= 1 - \exp\left[-\int_t^T \frac{\mathrm{d}\hat{t}}{\hat{t}} \int_{z_-}^{z_+} \mathrm{d}\hat{z}\, \frac{\alpha_s(\hat{t},\hat{z})}{2\pi} P_{\widetilde{ij}\to i,j}(\hat{z}) \mathrm{PDFR}_{\widetilde{ij}\to i,j}(\eta_{\widetilde{ij}},\hat{z},\hat{t})\right] \quad (2) \\
&= 1 - \exp\left[-\int_t^T \frac{\mathrm{d}\hat{t}}{\hat{t}} \mathcal{F}(\hat{t};\eta_{\widetilde{ij}})\right],
\end{aligned}
$$

---

[1]In the notation for backward evolution, we write $\widetilde{ij} \to i, j$ to mean a step where we have an $\widetilde{ij}$-flavoured parton at momentum fraction $\eta_{\widetilde{ij}}$ and consider the probability that it came from a parton of flavour $i$ at $\eta_i = \eta_{\widetilde{ij}}/\hat{z}$.

where $t$ is the evolution variable that evolves from $T$, eventually to a cutoff $t_c$, $z$ is the splitting variable used to define the fraction of the original momentum the emitter is left with after the emission, $\alpha_s$ is the strong coupling, $P$ is the splitting functions and $f$ are the PDFs, which are functions of the emitter's momentum fraction $\eta_{\tilde{ij}}$ as well as $t$. In the parton shower veto algorithm, we substitute the integrand $\mathcal{F}$ with an easy-to-integrate overestimate and use this to accept emissions,

$$\mathcal{G}(\hat{t}; \eta_{\tilde{ij}}) = \frac{1}{\hat{t}} \frac{\alpha_s^{\text{over}}}{2\pi} \text{PDFR}_{\tilde{ij} \to i,j}^{\max}(\eta_{\tilde{ij}}) \int_{z_-^{\text{over}}}^{z_+^{\text{over}}} \mathrm{d}\hat{z}\, P_{\tilde{ij} \to i,j}^{\text{over}}(\hat{z}) = \frac{1}{\hat{t}} c\,. \tag{3}$$

Compared to final state radiation, this overestimate contains the extra function $\text{PDF}_{\tilde{ij} \to i,j}^{\max}$. The rest of the formulation is unchanged.

## A.2 PDF Ratio Overestimates

Optimisation of the veto algorithm involves minimising the difference between the original components and their overestimate counterparts. For $\alpha_s$, the overestimate is $\alpha_s(t_c)$. For splitting functions, a simple overestimate function with the divergences can be defined. However, PDFs in Monte Carlo tools are not analytical functions – they are a grid of precisely measured values. Hence, the PDF ratio is a numerical function, and sampling often involves using the largest value observed from mass sampling. An improvement on this is to use the free variable $\eta_{\tilde{ij}}$ for a tighter overestimate. This optimisation is especially essential for $q \to g$ splitting, where the PDF ratio is not well behaved, and is often very large. This is because the quark PDFs are not well-behaved close to and below their nominal masses, and can be accounted for by changing the measure of the $\hat{t}$ integration,

$$\mathcal{P}_{\tilde{ij} \to i,j}(t, T; \eta_{\tilde{ij}}) \to 1 - \exp\left[-\int_t^T \frac{\mathrm{d}\hat{t}}{\hat{t} - m_q^2} \int_{z_-}^{z_+} \mathrm{d}\hat{z}\, \frac{\alpha_s(\hat{t}, \hat{z})}{2\pi} P_{\tilde{ij} \to i,j}(\hat{z}) \frac{\hat{t} - m_q^2}{\hat{t}} \frac{f(i, \eta_{\tilde{ij}}/\hat{z}, \hat{t})}{f(\tilde{ij}, \eta_{\tilde{ij}}, \hat{t})}\right],$$
$$\text{PDFR} \to \frac{\hat{t} - m_q^2}{\hat{t}} \frac{f(i, \eta_{\tilde{ij}}/\hat{z}, \hat{t})}{f(\tilde{ij}, \eta_{\tilde{ij}}, \hat{t})}\,. \tag{4}$$

We only perform this correction for $b$ and $c$ quarks, since the other quark masses are below the cutoff of the shower.

In our approach to obtain the PDF ratio overestimates, we started by setting it to 10,000 and running the simulation. For every PDF ratio calculation, we outputted the flavours $i$ and $\tilde{ij}$, $\eta_{\tilde{ij}}$ and PDFR. We then made a heatmap of PDFR against $\eta_{\tilde{ij}}$ to understand where the majority of the points lay. For most transitions, the top of the heatmap plateaus at a fixed value of the PDF ratio up to $\eta_{\tilde{ij}} \sim 0.1$, followed by higher values $\eta_{\tilde{ij}} \in [0.1, 1]$. From this observation, we used the sampled values to define the overestimate function based on the 99th and 99.99th percentile values,

$$\text{PDFR}_{\tilde{ij} \to i,j}^{\max}(\eta_{\tilde{ij}}) = \max(\text{PDFR}^{99\%}, \text{PDFR}^{99.99\%} \cdot \eta_{\tilde{ij}})\,. \tag{5}$$

The only exceptions to this fit were the $g \to u$ and $g \to d$ transitions, where we used

$$\text{PDFR}_{\tilde{ij} \to i,j}^{\max}(\eta_{\tilde{ij}}) = \text{PDFR}^{99.99\%} \cdot \sqrt{\eta_{\tilde{ij}}}\,. \tag{6}$$

Examples of heatmaps and their overestimate functions are presented in Figure 7.

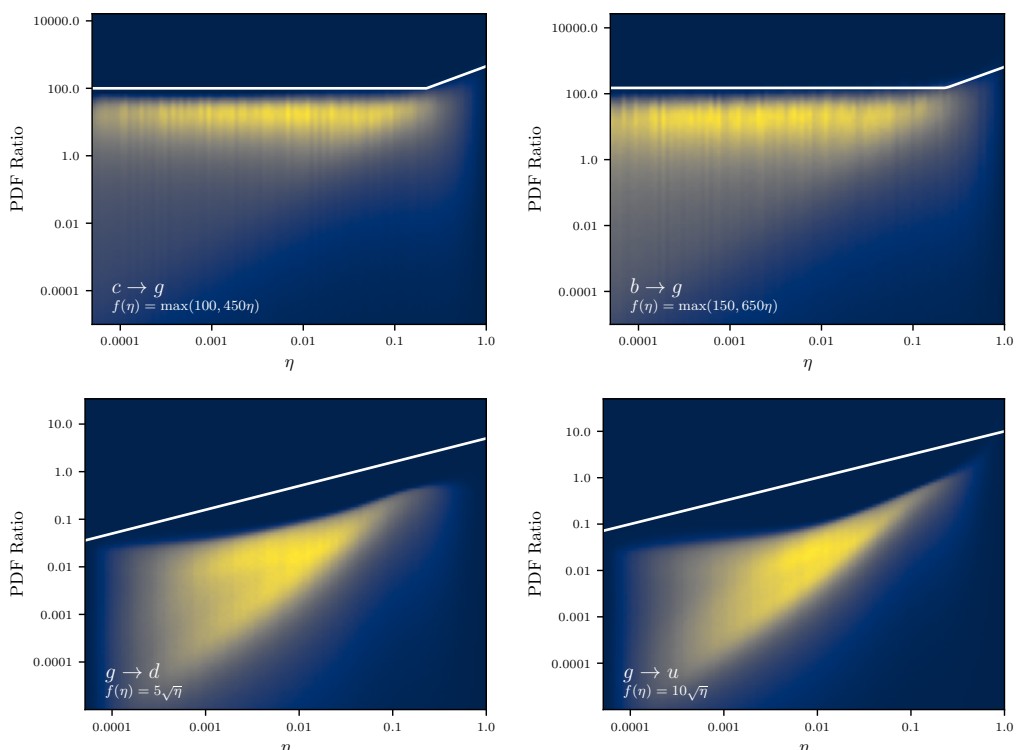

Figure 7: PDF Ratio Results for $c \to g$, $b \to g$, $g \to d$ and $g \to u$ splittings. The CT14lo set was used for the simulations. The chosen process was Z production at NLO, to incorporate the phase space for the power shower. The results show that the fitted equation overestimates the majority of the data points. The code to generate the data for this can be found in the C++ implementation, and the plotting code is in `plot-pdfratio.py`.

## B Appendix: Dipole Shower Prescription

The "classic" parton showers were designed to simulate $1 \to 2$ splittings. The first implementation of a dipole/antenna shower was in the ARIADNE Program [38], which generated final state $2 \to 3$ splittings. After twenty years, this paradigm of showers was revisited [39–41], this time based on the Kernels and Kinematics from Catani-Seymour Dipole Subtraction [42]. Inside modern event generators, the first implementations included the CSShower module in the Sherpa Event Generator [41,43] and the Dipole Shower Module in the Herwig Event Generator [25,44]. The transverse momentum ordered shower in the Pythia Event Generator was also modified to allow dipole-like splitting and recoil [45]. Modern parton shower research involves adjusting these showers to ensure next-to-leading-logarithmic accuracy [46–50].

### B.1 Framework and Convention

In our code, we will closely follow the Catani-Seymour prescription, as in Herwig. This section details the kernels, kinematics and sampling options implemented in our code. The dipole shower is designed to be ordered in transverse momentum,

$$t = p_\perp^2 = -k_\perp^\mu k_{\perp\mu}, \tag{7}$$

where $t$ is the ordering variable of the shower. The shower is designed to evolve the entire state, combining initial and final state emissions. The existence of dipoles with particles in the initial state or the final state leads to four types of splittings,

$$
\begin{aligned}
\text{Final}-\text{Final}\,(\text{FF}): \quad & \widetilde{ij} + \tilde{k} \rightarrow i + j + k\,, \\
\text{Final}-\text{Initial}\,(\text{FI}): \quad & \widetilde{ij} + \tilde{a} \rightarrow i + j + a\,, \\
\text{Initial}-\text{Final}\,(\text{IF}): \quad & \widetilde{aj} + \tilde{k} \rightarrow a + j + k\,, \\
\text{Initial}-\text{Initial}\,(\text{II}): \quad & \widetilde{aj} + \tilde{b} \rightarrow a + j + b\,,
\end{aligned}
\tag{8}
$$

where, in each case, the first particle on the left is split to the first two particles on the right, with the other particle as a spectator whose flavour is unchanged, but in most cases whose momentum is changed. The following sections will cover each case, highlighting the switch from the Catani-Seymour subtraction variables to shower variables and specifying the splitting kernels and kinematic mappings.

## B.2 Final-Final (FF) Splittings

Two variables define FF Splittings,

$$
z = \frac{p_i p_k}{p_i p_k + p_j p_k}\,, \quad y = \frac{p_i p_j}{p_i p_j + p_i p_k + p_j p_k}\,,
\tag{9}
$$

with $z, y \in [0, 1]$. These variables are used to define the splitting kernels

$$
\begin{aligned}
P_{q\rightarrow qg}^{\text{FF}}(z, y) &= C_F\left(\frac{2}{1 - z(1-y)} - (1+z)\right), \\
P_{g\rightarrow gg}^{\text{FF}}(z, y) &= \frac{C_A}{2}\left(\frac{2}{1 - z(1-y)} - 2 + z(1-z)\right), \\
P_{g\rightarrow q\bar{q}}^{\text{FF}}(z, y) &= \frac{T_R}{2}(1 - 2z(1-z)),
\end{aligned}
\tag{10}
$$

and the kinematics

$$
\begin{aligned}
p_i &= z & \tilde{p}_{ij} & + & y(1-z) & \tilde{p}_k & + & k_\perp\,, \\
p_j &= (1-z) & \tilde{p}_{ij} & + & yz & \tilde{p}_k & - & k_\perp\,, \\
p_k &= & & + & (1-y) & \tilde{p}_k\,. & &
\end{aligned}
\tag{11}
$$

The transverse momentum is given by

$$
t = z(1-z)y \cdot s_{\widetilde{ij},\tilde{k}}\,,
\tag{12}
$$

where $s_{ab} = 2p_a p_b$. Whilst $z$ can be used as the splitting variable, $y$ is replaced by $t$,

$$
\frac{\mathrm{d}y}{\mathrm{d}t} = \frac{y}{t} \rightarrow \frac{\mathrm{d}y}{y} = \frac{\mathrm{d}t}{t}\,.
\tag{13}
$$

As seen in [41], one can adjust the cross section after one emission, as given in [42, Eq. 5.20-27] as

$$
\begin{aligned}
\mathrm{d}\sigma_{n+1} &= \mathrm{d}\sigma_n \frac{\mathrm{d}y}{y} \mathrm{d}z \frac{\mathrm{d}\phi}{2\pi}(1-y)P(z, y) \\
&= \mathrm{d}\sigma_n \frac{\mathrm{d}t}{t} \mathrm{d}z \frac{\mathrm{d}\phi}{2\pi}(1-y)P(z, y)\,,
\end{aligned}
\tag{14}
$$

leading to the Sudakov Form Factor

$$\Delta^{\text{FF}} = \exp\left(-\int_{t_c}^{t_{\max}} \frac{\mathrm{d}t}{t} \int_{z_-}^{z_+} \mathrm{d}z \frac{\alpha_s(t)}{2\pi}(1-y)P(z,y)\right), \quad z_\pm = \frac{1}{2}\left(1 \pm \sqrt{1-\frac{4t}{s_{\tilde{ij},\tilde{k}}}}\right), \quad (15)$$

where $t_c$ is the shower cutoff. $t_{\max}$ can be set by the scale of the hard process, or by a previously generated or vetoed emission, or is otherwise equal to the kinematic limit,

$$t < \frac{s_{\tilde{ij},\tilde{k}}}{4}. \quad (16)$$

### B.3 Final-Initial (FI) Splittings

In FI splittings, $y$ is replaced with $x$,

$$z = \frac{p_i p_a}{p_i p_a + p_j p_a}, \quad x = \frac{p_i p_a + p_j p_a - p_i p_j}{p_i p_a + p_j p_a}, \quad (17)$$

with $x, z \in [0,1]$. The kernels are defined as

$$P_{q \to qg}^{\text{FI}}(z,x) = C_F\left(\frac{2}{1-z+(1-x)} - (1+z)\right),$$

$$P_{g \to gg}^{\text{FI}}(z,x) = \frac{C_A}{2}\left(\frac{2}{1-z+(1-x)} - 2 + z(1-z)\right), \quad (18)$$

$$P_{g \to q\bar{q}}^{\text{FI}}(z,x) = \frac{T_R}{2}(1 - 2z(1-z)),$$

and the kinematics are defined as

$$
\begin{aligned}
p_i &= z & \tilde{p}_{ij} &+ & (1-z)\frac{1-x}{x} & \tilde{p}_a &+ & k_\perp, \\
p_j &= (1-z) & \tilde{p}_{ij} &+ & z\frac{1-x}{x} & \tilde{p}_a &- & k_\perp, \\
p_a &= & & & \frac{1}{x} & \tilde{p}_a.
\end{aligned}
\quad (19)
$$

The transverse momentum is given by

$$t = z(1-z)\frac{1-x}{x} \cdot s_{\tilde{ij},\tilde{a}} \quad (20)$$

leading to the change of variables

$$\frac{\mathrm{d}x}{x} = (1-x)\frac{\mathrm{d}t}{t} \quad (21)$$

and the Sudakov Form Factor

$$\Delta^{\text{FI}} = \exp\left(-\int_{t_c}^{t_{\max}} \frac{\mathrm{d}t}{t} \int_{z_-}^{z_+} \mathrm{d}z \frac{\alpha_s(t)}{2\pi} P(z,x)\frac{f_a(\eta_a/x,t)}{f_a(\eta_a,t)}\right), \quad z_\pm = \frac{1}{2}\left(1 \pm \sqrt{1 - \frac{4t}{s_{\tilde{ij},\tilde{a}}}\frac{\eta_a}{1-\eta_a}}\right). \quad (22)$$

The limits on $z$ ensure that $x > \eta_a$. The kinematic limit is

$$t < \frac{s_{\tilde{ij},\tilde{a}}}{4}\frac{1-\eta_a}{\eta_a}. \quad (23)$$

## B.4 Initial-Final (IF) Splittings

In the initial state radiation case in Herwig, the backwards evolution is generated by sampling forward kinematic variables. In practice, the variables in the IF and II splittings need to be translated into $t$ and $z$. Further explanations are provided in [44] (the paper also defines a different kinematic mapping to the Catani-Seymour mapping, but Herwig provides a setting to switch between the two; we will use the Catani-Seymour mapping). IF splittings are defined using the variables

$$x = \frac{p_a p_j + p_a p_k - p_j p_k}{p_a p_j + p_a p_k}, \quad u = \frac{p_a p_j}{p_a p_j + p_a p_k}, \tag{24}$$

with $x, u \in [0, 1]$ and $x > \eta_{\tilde{a}j}$, the momentum fraction of the emitter. The kernels are defined as

$$\begin{aligned}
P^{\text{IF}}_{q \to qg}(x, u) &= C_F \left( \frac{2}{1 - x + u} - (1 + x) \right), \\
P^{\text{IF}}_{q \to g\bar{q}}(x, u) &= T_R (1 - 2x(1 - x)), \\
P^{\text{IF}}_{g \to gg}(x, u) &= C_A \left( \frac{1}{1 - x + u} + \frac{1 - x}{x} - 1 + x(1 - x) \right), \\
P^{\text{IF}}_{g \to qq}(x, u) &= C_F \left( x + 2 \frac{1 - x}{x} \right),
\end{aligned} \tag{25}$$

where, for example, $g \to qq$ implies the evolution of a gluon back to a quark by emission of a quark and, in all cases, the equivalent splitting functions under charge conjugation are equal. The kinematics are a mirror of the FI splitting, and are defined by

$$\begin{aligned}
p_a &= \frac{1}{x} & \tilde{p}_{aj}, \\
p_j &= (1 - u) \frac{1 - x}{x} & \tilde{p}_{aj} &+ u & \tilde{p}_k &- k_\perp, \\
p_k &= u \frac{1 - x}{x} & \tilde{p}_{aj} &+ (1 - u) & \tilde{p}_k &+ k_\perp.
\end{aligned} \tag{26}$$

For backward evolution of initial-state radiation, Herwig generates the forward evolution momentum fraction,

$$z = \frac{(p_a - p_j) \cdot \tilde{p}_k}{p_a \cdot \tilde{p}_k} = 1 - (1 - x)(1 - u) \tag{27}$$

and transverse momentum

$$t = \frac{u(1 - u)(1 - x)}{x} s_{\tilde{a}j, \tilde{k}}, \tag{28}$$

which are linked to the Catani-Seymour variables as

$$x = \frac{1 - z + r}{2r} \left[ 1 - \sqrt{1 - \frac{4rz(1 - z)}{(1 - z + r)^2}} \right], \quad u = \frac{xr}{1 - z}, \quad r = \frac{t}{s_{\tilde{a}j, \tilde{k}}}, \tag{29}$$

$$\frac{dx}{x} \frac{du}{u} = \frac{1}{u + x - 2ux} \frac{dt}{t} dz$$

giving the Sudakov Form Factor

$$\Delta^{\text{IF}} = \exp \left( - \int_{t_c}^{t_{\max}} \frac{dt}{t} \int_{z_-}^{z_+} dz \frac{\alpha_s(t)}{2\pi} \frac{1}{u + x - 2ux} P(x, u) \frac{f_a(\eta_{\tilde{a}j}/x, t)}{f_{\tilde{a}j}(\eta_{\tilde{a}j}, t)} \right),$$

$$z_\pm = \frac{1}{2} \left( (1 + \eta_{\tilde{a}j}) \pm (1 - \eta_{\tilde{a}j}) \sqrt{1 - \frac{4t\eta_{\tilde{a}j}}{s_{\tilde{a}j, \tilde{k}}(1 - \eta_{\tilde{a}j})}} \right). \tag{30}$$

The kinematic limit is again

$$t < \frac{s_{\tilde{a}j,\tilde{k}}}{4} \frac{1 - \eta_{\tilde{a}j}}{\eta_{\tilde{a}j}}. \tag{31}$$

## B.5 Initial-Initial (II) Splittings

II splittings are defined using the variables

$$x = \frac{p_a p_b - p_a p_j - p_b p_j}{p_a p_b}, \quad v = \frac{p_a p_j}{p_a p_b}, \tag{32}$$

with $x, v \in [0, 1]$ and $x > \eta_{\tilde{a}j}$, as in IF. The kernels are defined as

$$
\begin{aligned}
P^{\mathrm{II}}_{q \to qg}(x) &= C_F \left( \frac{2}{1-x} - (1+x) \right), \\
P^{\mathrm{II}}_{q \to g\bar{q}}(x) &= T_R (1 - 2x(1-x)), \\
P^{\mathrm{II}}_{g \to gg}(x) &= C_A \left( \frac{1}{1-x} + \frac{1-x}{x} - 1 + x(1-x) \right), \\
P^{\mathrm{II}}_{g \to qq}(x) &= C_F \left( x + 2\frac{1-x}{x} \right),
\end{aligned}
\tag{33}
$$

and the kinematics are defined as

$$
\begin{aligned}
p_a &= \frac{1}{x} && \tilde{p}_{aj}, \\
p_j &= \frac{1-x-v}{x} && \tilde{p}_{aj} \quad + \quad v \quad \tilde{p}_b \quad - \quad k_\perp, \\
p_b &= && \tilde{p}_b.
\end{aligned}
\tag{34}
$$

The imbalance in the total momentum is treated by giving a "kick" with a Lorentz Boost to all other final-state particles (not including the emission), given by

$$\Lambda^\mu_\nu = g^\mu_\nu - 2\frac{(K'+K)^\mu (K'+K)_\nu}{(K'+K)^2} + 2\frac{K^\mu K'_\nu}{K'^2} \tag{35}$$

$$K' = \tilde{p}_{aj} + \tilde{p}_b, \quad K = p_a + p_b - p_j$$

Herwig again generates the forward evolution momentum fraction, $z$ and transverse momentum squared, $t$,

$$z = \frac{(p_a - p_j) \cdot \tilde{p}_b}{p_a \cdot \tilde{p}_b} = x + v, \qquad t = \frac{v(1-x-v)}{x} s_{\tilde{a}j,\tilde{b}}, \tag{36}$$

which are linked to the Catani-Seymour variables as

$$x = \frac{z(1-z)}{1-z+r}, \quad v = \frac{xr}{1-z}, \quad r = \frac{t}{s_{\tilde{a}j,\tilde{b}}}, \quad \frac{dx}{x} \frac{dv}{v} = \frac{1}{z} \frac{dt}{t} dz, \tag{37}$$

giving the Sudakov Form Factor

$$\Delta^{\mathrm{II}} = \exp\left( -\int_{t_c}^{t_{\max}} \frac{dt}{t} \int_{z_-}^{z_+} dz \frac{\alpha_s(t)}{2\pi} \frac{1}{z} P(x) \frac{f_a(\eta_{\tilde{a}j}/x, t)}{f_{\tilde{a}j}(\eta_{\tilde{a}j}, t)} \right),$$

$$z_\pm = \frac{1}{2}\left( (1 + \eta_{\tilde{a}j}) \pm (1 - \eta_{\tilde{a}j}) \sqrt{1 - \frac{4t\eta_{\tilde{a}j}}{s_{\tilde{a}j,\tilde{b}}(1-\eta_{\tilde{a}j})^2}} \right). \tag{38}$$

The kinematic limit is again

$$t < \frac{s_{\tilde{a}j,\tilde{b}}}{4} \frac{1 - \eta_{\tilde{a}j}}{\eta_{\tilde{a}j}}. \tag{39}$$

## B.6 Deviations between Herwig and GAPS

There are a few differences in the implementation of the parton shower, as well as the event generator structure in Herwig, where reproducing those in GAPS was beyond the scope of this report.

In the Sudakov Form Factors for IF and II splittings, the Jacobian Factor diverges for specific values of $x, u/v$. In this case, either a very large overestimate is required to handle the size of this, or advanced overestimation must be employed. In Herwig, the overestimate is generated using adaptive sampling through the ExSample package [51], which is a better option than the large overestimates employed in our code. However, employing adaptive overestimation is beyond the scope of this study and is a starting point for future developments. Additionally, future shower projects should employ mathematical alternatives to avoid divergences in the Sudakov Form Factor.

When it comes to PDF Evaluation, Herwig applies extrapolation of the PDFs at large $x$, and also uses a freezing scale for the $Q^2$ value going into the PDF. These two settings are not adjustable by the user. For this study, we manually edited the code to ensure that there is no $x$ extrapolation and the PDF evaluates to 0 outside the $Q^2$ limits, as is in our code. One can adjust the Herwig code with the file `herwig-for-gaps.diff` provided in the codebase.

Outside the parton shower, events in Herwig also undergo beam remnant generation. After the event has been showered, Herwig ensures that the two initial partons are valence quarks, such that the beam remnants can be cast into diquarks. This is achieved by generating forced splitting of the initial partons. Additionally, all final gluons are also split into quarks by this mechanism. We minimise the effect of this in our analysis by omitting partons whose parents were beam remnants. However, the changes in momenta due to the forced splitting were unavoidable, as the remnant decay mechanism is deeply integrated in the Herwig workflow and cannot be turned off. [52].

# C Appendix: NLO Calculation and Matching

In order to facilitate comparisons with existing implementations, for the NLO calculation and matching we simplify the process to on-shell $Z$ production, $pp \to Z$, rather than performing the entire process $pp \to Z\gamma \to e^+ e^-$. This allows us to implement exactly the derivation of NLO + CS Subtraction from [32, ch. 3.3], with two differences: their example studies $W$ production, whereas we produce $Z$, and that we include the case of $gq \to Zq$. We use results and notation from [31] for the matrix elements themselves. A future extension will be to the full $pp \to Z\gamma \to e^+ e^-$ process, for which the "Kleiss trick" [53] will be useful, since this writes all NLO matrix elements in terms of LO ones, and requires no additional electroweak calculation.

## C.1 Born Event

We label the particles and momenta as

$$a(p_a) + b(p_b) \to 1(p_1), \quad m_1 = m_z, \tag{40}$$

and use the spin and colour averaged squared matrix element for the process $q\bar{q} \to Z$ [31, ch. 8, 9],

$$\left|\overline{\mathcal{M}}_{q\bar{q}\to Z}\right|^2 = \frac{1}{N_c}\frac{1}{4}\sqrt{2}G_F m_1^4(V_q^2 + A_q^2). \tag{41}$$

The differential phase space is

$$\begin{aligned}
\mathrm{d}\Phi_1 &= \frac{\mathrm{d}^4 p_1}{(2\pi)^4} 2\pi\delta(p_1^2 - m_1^2)(2\pi)^4\delta^4(p_1 - (p_a + p_b)) \\
&= 2\pi\delta((p_a + p_b)^2 - m_1^2) \\
&= 2\pi\delta(\hat{s} - m_1^2),
\end{aligned} \tag{42}$$

and the partonic cross section is

$$\mathrm{d}\hat{\sigma}_{2\to 1} = \frac{1}{2\hat{s}}\left|\overline{\mathcal{M}}_{q\bar{q}\to Z}\right|^2 2\pi\delta(\hat{s} - m_1^2) = \frac{\pi}{\hat{s}}\left|\overline{\mathcal{M}}_{q\bar{q}\to Z}\right|^2 \delta(\hat{s} - m_1^2), \tag{43}$$

in agreement with [32, ch. 2.2]. This additional delta function constrains our integral over $\hat{s}$ in the hadronic cross section. Including PDFs, the cross section is given by

$$\begin{aligned}
\mathrm{d}\sigma_{2\to 1} &= 2\sum_{a,b}\frac{\mathrm{d}\hat{s}\,\mathrm{d}y}{\hat{s}}\cdot\eta_a f_a(\eta_a,\mu_F)\eta_b f_b(\eta_b,\mu_F)\frac{\pi}{\hat{s}}\left|\overline{\mathcal{M}}_{q\bar{q}\to Z}\right|^2\delta(\hat{s} - m_1^2) \\
&= 2\sum_{a,b}\pi\mathrm{d}y\cdot\eta_a f_a(\eta_a,\mu_F)\eta_b f_b(\eta_b,\mu_F)\left|\overline{\mathcal{M}}_{q\bar{q}\to Z}\right|^2\frac{1}{m_1^4} \\
&= 2\sum_{a,b}\pi\cdot\ln\frac{s}{m_1^2}\mathrm{d}\rho_1\cdot\eta_a f_a(\eta_a,\mu_F)\eta_b f_b(\eta_b,\mu_F)\left|\overline{\mathcal{M}}_{q\bar{q}\to Z}\right|^2\frac{1}{m_1^4},
\end{aligned} \tag{44}$$

where the sum is over the five flavours. The extra factor of 2 comes from the two possible orientations of the quark antiquark pair, $P(q)P(\bar{q})$ and $P(\bar{q})P(q)$. The momentum fractions $\eta_{1,2}$ are related to $\hat{s}$ and the $Z$ rapidity, $y$, by $\eta_{1,2} = \mathrm{e}^{\pm y}\sqrt{\hat{s}/s}$. The final line shows the mapping to a uniform random number on the range $(0,1)$, $\rho_1$, used to generate the phase space point.

## C.2 Real Emission

We have three processes to simulate,

$$\begin{aligned}
q(p_a) + \bar{q}(p_b) &\to Z(p_1) + g(p_2), \\
g(p_a) + q(p_b) &\to Z(p_1) + q'(p_2), \\
g(p_a) + \bar{q}(p_b) &\to Z(p_1) + \bar{q}'(p_2),
\end{aligned} \tag{45}$$

To simplify our formulae, we can define new Mandelstam variables with the momenta of the quark, antiquark, $Z$ and gluon,

$$\begin{aligned}
\bar{s} &= (p_a + p_b)^2 = (p_1 + p_2)^2, \\
\bar{t} &= (p_a - p_1)^2 = (p_b - p_2)^2, \\
\bar{u} &= (p_a - p_2)^2 = (p_b - p_1)^2.
\end{aligned} \tag{46}$$

We use the bar notation to differentiate between S (Born) ($\hat{s}$) and H (real emission) ($\bar{s}$) events. We define the kinematics of the real emission event from those of a Born event using the same

variables as if it was an initial-initial parton shower emission, see Eqs. (32, 34, 35). In our case, $j = 2$. The NLO Matrix Elements can be obtained from the one for $e^+e^-$ annihilation [31, ch. 7],

$$\left|\overline{\mathcal{M}}_{q\bar{q}\to Zg}\right|^2 = \pi\alpha_s\sqrt{2}G_F m_1^2(V_q^2 + A_q^2)\frac{2C_F}{N_c}\left(\frac{\bar{t}}{\bar{u}} + \frac{\bar{u}}{\bar{t}} + \frac{2m_1^2\bar{s}}{\bar{t}\bar{u}}\right)$$
$$= \frac{8\pi\alpha_s C_F}{m_1^2}\left|\overline{\mathcal{M}}_{q\bar{q}\to Z}\right|^2\left(\frac{\bar{t}}{\bar{u}} + \frac{\bar{u}}{\bar{t}} + \frac{2m_1^2\bar{s}}{\bar{t}\bar{u}}\right),$$
$$\left|\overline{\mathcal{M}}_{gq\to Zq}\right|^2 = \pi\alpha_s\sqrt{2}G_F m_1^2(V_q^2 + A_q^2)\frac{2T_R}{N_C}\left(-\frac{\bar{u}}{\bar{s}} - \frac{\bar{s}}{\bar{u}} - \frac{2m_1^2\bar{t}}{\bar{s}\bar{u}}\right)$$
$$= \frac{8\pi\alpha_s T_R}{m_1^2}\left|\overline{\mathcal{M}}_{q\bar{q}\to Z}\right|^2\left(-\frac{\bar{u}}{\bar{s}} - \frac{\bar{s}}{\bar{u}} - \frac{2m_1^2\bar{t}}{\bar{s}\bar{u}}\right).$$
(47)

Since we want to match the real emission with our dipole shower, we have to subtract the dipole shower distribution from it. And since our dipole shower is based on the CS kinematics and distributions, our subtraction terms are exactly those of the CS subtraction algorithm. We can make the subtraction easier by manipulating the matrix element expressions into forms in which the CS subtraction terms are manifest. For this we use the definition of $x$, which, in our scenario, is

$$x = \frac{m_1^2}{\bar{s}}, \quad 1 - x = \frac{-(\bar{t} + \bar{u})}{\bar{s}}, \quad 1 + x = \frac{m_1^2 + s}{s}.$$
(48)

Starting with the gluon emission case, we can follow the example of [32, ch. 3.3],

$$\frac{\bar{t}}{\bar{u}} + \frac{\bar{u}}{\bar{t}} + \frac{2m_1^2\bar{s}}{\bar{t}\bar{u}} = -\bar{s}\left(\frac{1}{\bar{t}} + \frac{1}{\bar{u}}\right)\left(-\frac{2\bar{s}}{\bar{t} + \bar{u}} - \frac{m_1^2 + \bar{s}}{\bar{s}}\right) - 2$$
$$= -\frac{m_1^2}{x}\left(\frac{1}{\bar{t}} + \frac{1}{\bar{u}}\right)\left(\frac{2}{1-x} - (1+x)\right) - 2.$$
(49)

Next, for the initial state gluon case,

$$-\frac{\bar{u}}{\bar{s}} - \frac{\bar{s}}{\bar{u}} - \frac{2m_1^2\bar{t}}{\bar{s}\bar{u}} = -\frac{1}{\bar{u}}\left(\bar{s} + \frac{2m_1^2(\bar{t} + \bar{u})}{\bar{s}}\right) - \frac{\bar{u} - 2m_1^2}{\bar{s}}$$
$$= -\frac{m_1^2}{x}\frac{1}{\bar{u}}\left(1 - 2x(1-x)\right) + \frac{2m_1^2 - \bar{u}}{\bar{s}},$$
(50)

Then we calculate our dipole shower subtraction terms in the same variables,

$$\mathcal{D}_{q\to qg} = 8\pi\alpha_s C_F\left|\overline{\mathcal{M}}_{q\bar{q}\to Z}\right|^2\frac{1}{x}\frac{1}{\bar{u}}\left(\frac{2}{1-x} - (1+x)\right),$$
$$\mathcal{D}_{q\to g\bar{q}} = 8\pi\alpha_s T_R\left|\overline{\mathcal{M}}_{q\bar{q}\to Z}\right|^2\frac{1}{x}\frac{1}{\bar{u}}\left(1 - 2x(1-x)\right).$$
(51)

For gluon emission, both $\mathcal{D}_{q\to qg}$ and an analogous $\mathcal{D}_{\bar{q}\to\bar{q}g}$ term with $\bar{u} \to \bar{t}$ need to be subtracted, whereas for initial-state gluons, $\mathcal{D}_{q\to g\bar{q}}$ and the analogous $\mathcal{D}_{\bar{q}\to gq}$ term are subtracted from different processes. In the case that our shower covers the whole of phase space, then the subtraction terms are subtracted from the real emission everywhere in phase space, and we obtain simple formulae for the difference,

$$(\mathcal{R} - \mathcal{S})^{q\bar{q}\to Zg} = \frac{8\pi\alpha_s C_F}{m_1^2}\left|\overline{\mathcal{M}}_{q\bar{q}\to Z}\right|^2\left(-2\right),$$
$$(\mathcal{R} - \mathcal{S})^{gq\to Zq} = \frac{8\pi\alpha_s T_R}{m_1^2}\left|\overline{\mathcal{M}}_{q\bar{q}\to Z}\right|^2\left(\frac{2m_1^2 - \bar{u}}{\bar{s}}\right).$$
(52)

Lastly, we need to consider the cross section formula. The differential cross section can be written as

$$
\begin{aligned}
\mathrm{d}\sigma_{pp\to 12} &= 2\sum_{a,b} \frac{\mathrm{d}\eta_a}{\eta_a}\frac{\mathrm{d}\eta_b}{\eta_b} \cdot \eta_a f_a \eta_b f_b \cdot \mathrm{d}\hat{\sigma}_{ab\to 12} \\
&= 2\sum_{a,b} \frac{\mathrm{d}\bar{s}\,\mathrm{d}y}{\bar{s}} \cdot \eta_a f_a \eta_b f_b \cdot \frac{1}{2\bar{s}}(\mathcal{R}-\mathcal{S})\mathrm{d}\Phi_2 \\
&= 2\sum_{a,b} \mathrm{d}y \cdot \eta_a f_a \eta_b f_b \cdot (\mathcal{R}-\mathcal{S})\frac{1}{16\pi}\frac{\mathrm{d}\phi}{2\pi}\mathrm{d}x\mathrm{d}\nu\frac{1}{m_1^2},
\end{aligned}
\tag{53}
$$

where $y$ is now the rapidity of the hard system and $\eta_{1,2} = \mathrm{e}^{\pm y}\sqrt{\bar{s}/s}$.

## C.3   Born + Virtual + Insertion + Collinear

The interference between the 1-loop and tree-level matrix elements, which we call the virtual contribution and write symbolically as $\left|\overline{\mathcal{M}}\right|^2_{loop}$, is given by [32, ch. 2]

$$
\left|\overline{\mathcal{M}}_{q\bar{q}\to Z}\right|^2_{loop} = C_F\frac{\alpha_s}{2\pi}\left|\overline{\mathcal{M}}_{q\bar{q}\to Z}\right|^2 \frac{1}{\Gamma(1-\epsilon)}\left(\frac{4\pi\mu^2}{Q^2}\right)^\epsilon\left[-\frac{2}{\epsilon^2}-\frac{3}{\epsilon}-8+\pi^2\right].
\tag{54}
$$

According to the Catani-Seymour algorithm, its singularities are cancelled by insertion operators that are calculated as integrals of the emission kernels for the quark and the antiquark. Together, they form the term

$$
I = C_F\frac{\alpha_s}{2\pi}\left|\overline{\mathcal{M}}_{q\bar{q}\to Z}\right|^2\frac{1}{\Gamma(1-\epsilon)}\left(\frac{4\pi\mu^2}{Q^2}\right)^\epsilon\left[+\frac{2}{\epsilon^2}+\frac{3}{\epsilon}+10-\pi^2\right],
\tag{55}
$$

which leaves a factor of 2 inside the brackets. As the phase space is left identical, this factor can be added to the differential cross section.

$$
\mathrm{d}\sigma^{\mathcal{B}+\mathcal{V}+\mathcal{I}} = \mathrm{d}\sigma^{\mathcal{B}}\left(1 + C_F\frac{\alpha_s}{2\pi}(2)\right).
\tag{56}
$$

This term numerically approximates to $\mathrm{d}\sigma^{\mathcal{B}}\times 1.05$, only a small correction.

Next, we look at the Collinear terms and their convolution with the PDFs. We can start by looking at the differential cross section of this component,

$$
\begin{aligned}
\mathrm{d}\sigma^{\mathcal{C}} = &\sum_{a,b,a',b'}\int \mathrm{d}x\frac{\mathrm{d}\eta_a}{\eta_a}\frac{\mathrm{d}\eta_b}{\eta_b} \cdot \eta_a f_a \eta_b f_b \cdot \mathrm{d}\hat{\sigma}^{\mathcal{B}}_{a'b'}x\eta_a P_a, \eta_b P_b)\cdot(\mathcal{K}^{a,a'}(x)+\mathcal{P}^{a,a'}(x))\delta^{b,b'}, \\
+ &\sum_{a,b,a',b'}\int \mathrm{d}x\frac{\mathrm{d}\eta_a}{\eta_a}\frac{\mathrm{d}\eta_b}{\eta_b} \cdot \eta_a f_a \eta_b f_b \cdot \mathrm{d}\hat{\sigma}^{\mathcal{B}}_{a'b'}(\eta_a P_a, x\eta_b P_b)\cdot(\mathcal{K}^{b,b'}(x)+\mathcal{P}^{b,b'}(x))\delta^{a,a'},
\end{aligned}
\tag{57}
$$

where $P_i$ are proton momenta. Let us work with the quark emitter case for now, as the antiquark emitter case is identically studied. We can manipulate this integral over $x$ by inserting the integral $\int \mathrm{d}\bar{\eta}_a\delta(\bar{\eta}_a - x\eta_a)$,

$$
\int \frac{\mathrm{d}\eta_a}{\eta_a}\mathrm{d}\bar{\eta}_a\delta(\bar{\eta}_a - x\eta_a) = \int \frac{\mathrm{d}\bar{\eta}_a}{\bar{\eta}_a},
\tag{58}
$$

allowing us to make the switch $\eta_a \to \bar{\eta}_a/x$ and similarly, $\eta_b \to \bar{\eta}_b/x$ in the second integral. Note that, to avoid the momentum fraction becoming larger than 1, this comes with a physical boundary,

$$
\Theta(x - \bar{\eta}_a).
\tag{59}
$$

As $\mathrm{d}\eta_a/\eta_a = \mathrm{d}\bar{\eta}_a/\bar{\eta}_a$, we can manipulate our integrals to factor out the Born differential cross section,

$$
\mathrm{d}\sigma_a^{\mathcal{C}} = \sum_{a,b,a',b'} \int \mathrm{d}x \frac{\mathrm{d}\bar{\eta}_a}{\bar{\eta}_a} \frac{\mathrm{d}\eta_b}{\eta_b} \cdot \bar{\eta}_a f_{a'}(\bar{\eta}_a) \eta_b f_{b'} \cdot \mathrm{d}\hat{\sigma}_{a'b'}^{\mathcal{B}}(\bar{\eta}_a P_a, \eta_b P_b) \cdot (\mathcal{K}^{a,a'} + \mathcal{P}^{a,a'}) \frac{\frac{\bar{\eta}_a}{x} f_a(\frac{\bar{\eta}_a}{x})}{\bar{\eta}_a f_{a'}(\bar{\eta}_a)} \delta^{b,b'}
$$

$$
= \mathrm{d}\sigma^{\mathcal{B}} \sum_a \int \mathrm{d}x (\mathcal{K}^{a,a'} + \mathcal{P}^{a,a'}) \frac{\frac{\bar{\eta}_a}{x} f_a(\frac{\bar{\eta}_a}{x})}{\bar{\eta}_a f_{a'}(\bar{\eta}_a)},
$$

$$
\mathrm{d}\sigma^{\mathcal{C}} = \mathrm{d}\sigma^{\mathcal{B}} \left( \sum_a \int \mathrm{d}x (\mathcal{K}^{a,a'} + \mathcal{P}^{a,a'}) \frac{\frac{\bar{\eta}_a}{x} f_a(\frac{\bar{\eta}_a}{x})}{\bar{\eta}_a f_{a'}(\bar{\eta})} + \sum_b \int \mathrm{d}x (\mathcal{K}^{b,b'} + \mathcal{P}^{b,b'}) \frac{\frac{\bar{\eta}_b}{x} f_b(\frac{\bar{\eta}_b}{x})}{\bar{\eta}_b f_{b'}(\bar{\eta}_b)} \right).
$$
(60)

This allows us to add another contribution inside the brackets of (56). Here, we note down all the required $\mathcal{K}$ and $\mathcal{P}$ terms for the case without any final state partons,

$$
\mathcal{K}^{q,q} = \frac{C_F \alpha_s(\mu_R)}{2\pi} \left( \bar{\mathcal{K}}^{qq} + \tilde{\mathcal{K}}^{qq} \right),
$$

$$
\bar{\mathcal{K}}^{qq} = \left( \frac{2}{1-x} \ln\left( \frac{1-x}{x} \right) \right)_+ - (1+x) \ln\left( \frac{1-x}{x} \right) + (1-x) - (5 - \pi^2)\delta(1-x),
$$

$$
\tilde{\mathcal{K}}^{qq} = \left( \frac{2}{1-x} \ln(1-x) \right)_+ - (1+x) \ln(1-x) - \frac{\pi^2}{3}\delta(1-x),
$$

$$
\mathcal{P}^{q,q} = -\frac{C_F \alpha_s(\mu_R)}{2\pi} \ln\left( \frac{\mu_F^2}{m_1^2} \right) \left( \left( \frac{2}{1-x} \right)_+ - (1+x) + \frac{3}{2}\delta(1-x) \right),
$$
(61)

$$
\mathcal{K}^{g,q} = \frac{T_R \alpha_s(\mu_R)}{2\pi} \left( \bar{\mathcal{K}}^{gq} + \tilde{\mathcal{K}}^{gq} \right),
$$

$$
\bar{\mathcal{K}}^{gq} = (x^2 + (1-x)^2) \ln\left( \frac{1-x}{x} \right) + 2x(1-x),
$$

$$
\tilde{\mathcal{K}}^{gq} = (x^2 + (1-x)^2) \ln(1-x),
$$

$$
\mathcal{P}^{g,q} = -\frac{T_R \alpha_s(\mu_R)}{2\pi} \ln\left( \frac{\mu_F^2}{m_1^2} \right) (x^2 + (1-x)^2).
$$

We can combine these terms and group common terms. The terms, when fully written out, have the form

$$
\mathcal{K}(x) + \mathcal{P}(x) = g(x)_+ + h(x) + A\delta(1-x)
$$

$$
= g(x) - \delta(1-x) \left( \int \mathrm{d}x' g(x') \right) + h(x) + A\delta(1-x),
$$
(62)

We can insert this into our integral. Remembering our physical boundary $\Theta(x - \bar{\eta}_a)$, we add in the limits on the integrals,

$$
\int_0^1 \mathrm{d}x \frac{\frac{\bar{\eta}_a}{x} f_a(\frac{\bar{\eta}_a}{x})}{\bar{\eta}_a f_{a'}(\bar{\eta}_a)} (\mathcal{K}(x) + \mathcal{P}(x))
$$

$$
= \int_0^1 \mathrm{d}x \frac{\frac{\bar{\eta}_a}{x} f_a(\frac{\bar{\eta}_a}{x})}{\bar{\eta}_a f_{a'}(\bar{\eta}_a)} \left( g(x) - \delta(1-x) \left( \int_0^1 \mathrm{d}x' g(x') \right) + h(x) + A\delta(1-x) \right)
$$

$$
= A - \int_0^{\bar{\eta}_a} \mathrm{d}x g(x) - \int_{\bar{\eta}_a}^1 \mathrm{d}x g(x) + \int_{\bar{\eta}_a}^1 \mathrm{d}x \frac{\frac{\bar{\eta}_a}{x} f_a(\frac{\bar{\eta}_a}{x})}{\bar{\eta}_a f_{a'}(\bar{\eta}_a)} (g(x) + h(x))
$$
(63)

$$
= A - B(\bar{\eta}_a) + \int_{\bar{\eta}_a}^1 \mathrm{d}x \left( \left( \frac{\frac{\bar{\eta}_a}{x} f_a(\frac{\bar{\eta}_a}{x})}{\bar{\eta}_a f_{a'}(\bar{\eta}_a)} - 1 \right) g(x) + \frac{\frac{\bar{\eta}_a}{x} f_a(\frac{\bar{\eta}_a}{x})}{\bar{\eta}_a f_{a'}(\bar{\eta}_a)} h(x) \right).
$$

We can calculate the constant $A$ and function $B$ and sample over $x$ for the integral. Like the hard emission, we have four possible splittings. In this case, the terms are given by

$$
\begin{aligned}
A^{qq} &= C_F \frac{\alpha_s}{2\pi} \left( -5 + \frac{2\pi^2}{3} - \frac{3}{2} \ln\left( \frac{\mu_F^2}{m_1^2} \right) \right), \\
B^{qq} &= C_F \frac{\alpha_s}{2\pi} \left( 2\left( \ln(1-\bar{\eta}_a) \ln\left( \frac{\mu_F^2}{m_1^2} \right) + \frac{\pi^2}{6} - \ln^2(1-\bar{\eta}_a) - \mathrm{Li}_2(1-\bar{\eta}_a) \right) \right), \\
g^{qq}(x) &= C_F \frac{\alpha_s}{2\pi} \left( \frac{2}{1-x} \left( \ln\frac{1-x}{x} + \ln(1-x) - \ln\left( \frac{\mu_F^2}{m_1^2} \right) \right) \right), \\
A^{qg} &= B^{qg} = g^{qg}(x) = 0, \\
h^{qq}(x) &= C_F \frac{\alpha_s}{2\pi} \left( -(1+x)\left( \ln\frac{1-x}{x} + \ln(1-x) - \ln\left( \frac{\mu_F^2}{m_1^2} \right) \right) + (1-x) \right), \\
h^{qg}(x) &= T_R \frac{\alpha_s}{2\pi} \left( 2x(1-x) + (x^2 + (1-x)^2)\left( \ln\frac{1-x}{x} + \ln(1-x) - \ln\left( \frac{\mu_F^2}{m_1^2} \right) \right) \right).
\end{aligned}
\tag{64}
$$

We can combine all of our terms into a single variable,

$$
d\sigma^{\mathcal{C}} = d\sigma^{\mathcal{B}} C_F \frac{\alpha_s}{2\pi} \kappa_{\mathcal{C}}.
\tag{65}
$$

This gives the full Born + Virtual + Insertion + Collinear term

$$
d\sigma^{\mathcal{B}+\mathcal{V}+\mathcal{I}+\mathcal{C}} = d\sigma^{\mathcal{B}} \left( 1 + C_F \frac{\alpha_s}{2\pi}(2 + \kappa_{\mathcal{C}}) \right).
\tag{66}
$$

### C.4 Matching to the Parton Shower

Once the H and S events have been generated, they need to be showered. There are many possible ways to connect the shower to the NLO code, with the subtraction term in H events matching the showering of S events. We have chosen the simplest approach, where the ordering variable is set to the hadronic centre-of-mass energy squared, performing a "power shower" [54]. Alternative choices and their behaviour can be studied using Herwig [55]. As mentioned above, this means that the subtraction term applies everywhere in phase space, and we can write H events in terms of $(\mathcal{R} - \mathcal{S})$.

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
