# Peer review of "An NLO-Matched Initial and Final State Parton Shower on a GPU"

_SciPost Physics Codebases_

## Round 1 · Referee Report · Peter Skands (Referee 1) · 2026-1-5

Report

See requested changes.

Requested changes

For cross checks, it would be useful to be able to build and run the CPU and/or GPU codes independently of each other. The README files in the repository do not appear to contain instructions for how to do so. E.g., I do not have an NVIDIA GPU but would still have liked to be able to test that at least the CPU code compiles. This I could not do.

In section 2: on the discussion of selecting the winner. As discussed in more detail in 1605.09246, the veto algorithm can also be run using the sum of trial weights, in that paper called ‘generate-select’. On a CPU, this tends to be faster than the way discussed here especially when the number of emitters is large. Why is this not considered or commented on here? (This more efficient way of generating trials has already, e.g., been used in the Vincia parton shower. I believe it is also used in the PanScales showers, and perhaps others. )

Figure 1: is not clear / lacks explanation. What do the numbers in the boxes mean? What do green or red circles mean? What do dashed lines mean? What do dashed boxes mean? What do ellipses mean? What do vertical rows correspond to? What does “Cycle 1” refer to? The earlier paper [11] does contain more explanation but even there the precise meaning of all graphic elements is still not really clear enough in my opinion. And this paper should be readable without having to refer back to [11] anyway. I note, however, that once Fig 1 has been improved (by improving the explanation and/or by improving the figure) I think Fig 2 is good and clear.

In section 3.1: While it might reduce power consumption, it is not a priori clear to me that running the GPU with fewer kernels would SPEED UP the calculation; what processing effect results in the overall speedup seen from partitioning?

In section 4.1: define the observables mZ, pT, etc, in terms of how they are calculated from the event records. Using which particle(s)?

In section 4.1: It should be made more clear that in the unmatched case, the effect of the Z width is included, while in the NLO one it is not. I accept that this is implicit from the process specifications, but since the arrows in those specifications are not rigorously defined it could remain ambiguous which precise approximation they imply.

Figure 3: phi(Z) distribution. The spike in the middle looks like a misbinning effect? Likewise the drops at the edges. Likewise the phi(Z) plot in Figure 4.

Figure 3 caption: the phrasing "is shown when" is potentially confusing. Rephrase the statements describing the cuts for clarity and conciseness.

Figure 4, first panel: nice to see that the stated delta function is actually what is generated, but does this really require an entire plot?

Figure 4: what is the feature below pTZ ~ 1.5 GeV ? Presumably related to the cutoff, but I could not find any comment / explanation. Also, I note that this plot is in log pT, while the corresponding one in Fig 3 was on a linear scale, making it hard to compare.

Figure 5 caption: the statement "for small NEV there is negligible improvement" is misleading, and indeed negated by the following clause. Rephrase for correctness and conciseness.

Figure 6: here, I would like to see the extent to which linear scaling is or is not manifested. The choice of log x axis distorts that. Suggest to change to execution time PER fixed number of events and equivalently power consumption PER fixed number of events.

Figure 6: it is worthwhile and interesting to consider the energy consumption. Another factor would be the effective price in money. Are the prices to acquire these two systems similar? What about renting them from a service provider for the calculation? These numbers would be interesting to add.

The final paragraph contains too many colloquialisms, which would be fine in a seminar setting, but seem misplaced here.

In appendix A: I think it would be appropriate to cite the original derivation of the backwards-evolution framework for ISR,
https://doi.org/10.1016/0370-2693(85)90674-4

In Appendix B: Together with the first mention of the ARIADNE program, it would also be fair to include the original dipole-antenna shower paper it was based on,
https://doi.org/10.1016/0550-3213(88)90441-5.
And since a fairly comprehensive list of showers is given, I think it would be appropriate to mention that Pythia also implements an option for Ariadne-like antenna showers,
https://arxiv.org/abs/2003.00702.

In Appendix C: Clarify whether this is an “MC@NLO”-style matching, a POWHEG-style one, or something else, along with appropriate references?

Recommendation

Ask for major revision

---

## Editorial Decision

in_refereeing